# A Robust Method for Automatic Panoramic UAV Image Mosaic

**DOI:** 10.3390/s19081898

**Published:** 2019-04-22

**Authors:** Jun Chen, Quan Xu, Linbo Luo, Yongtao Wang, Shuchun Wang

**Affiliations:** 1School of Automation, China University of Geosciences, Wuhan 430074, China; chenjun71983@163.com (J.C.); ytwang@cug.edu.cn (Y.W.); wangscgt@126.com (S.W.); 2Hubei Key Laboratory of Advanced Control and Intelligent Automation for Complex Systems, Wuhan 430074, China; 3Hubei Provincial Key Laboratory of Intelligent Robot, Wuhan Institute of Technology, Wuhan 430073, China; 4School of Mechanical Engineering and Electronic Information, China University of Geosciences, Wuhan 430074, China; 1816312309@163.com

**Keywords:** image mosaic, nonrigid deformation, bundle adjustment, feature matching

## Abstract

This paper introduces a robust method for panoramic unmanned aerial vehicle (UAV) image mosaic. In the traditional automatic panoramic image stitching method (Autostitch), it assumes that the camera rotates about its optical centre and the group of transformations the source images may undergo is a special group of homographies. It is rare to get such ideal data in reality. In particular, remote sensing images obtained by UAV do not satisfy such an ideal situation, where the images may not be on a plane yet and even may suffer from nonrigid changes, leading to poor mosaic results. To overcome the above mentioned challenges, in this paper a nonrigid matching algorithm is introduced to the mosaic system to generate accurate feature matching on remote sensing images. We also propose a new strategy for bundle adjustment to make the mosaic system suitable for the UAV image panoramic mosaic effect. Experimental results show that our method outperforms the traditional method and some of the latest methods in terms of visual effect.

## 1. Introduction

Unmanned aerial vehicle (UAV) remote sensing is a low-altitude technology, which has become an important means of information acquisition. UAV has been widely used in various ground survey applications due to its advantages of high acquisition speed, convenient operation, good safety, and low investment. However, a UAV image has a small view of scene due to the low flying altitude and limited focal length of the camera; thus, capturing a relatively complete target area becomes difficult. Multiple images of the same target should be combined by technical means to obtain a complete scene of the desired target. The purpose of the image mosaic is to combine multiple images with overlapping fields to form a panoramic image [1,2,3].

Traditional image stitching [4] includes feature matching, image matching, bundle adjustment, automatic panorama straightening, gain compensation, and multiband blending [5,6]. This system may appear perfect, but several errors emerge when images contain a large amount of nonideal data. For example, when UAV images are stitched using this system, ghosting and even incorrect matches may arise. Several efforts have been exerted to reduce these errors, for example, seam cutting methods [7,8] optimize pixel selection among the overlapping images to minimize visible seams, whereas Laplacian pyramid blending [4,5] and Poisson image blending [9] minimize blurring due to misalignments or exposure differences. Nevertheless, the effect remains unsatisfactory, and the problem may be even further severe when scenes are captured by UAVs involving nonrigid changes.

Two major challenges exist in the traditional stitching system when applying to UAV image mosaics. On the one hand, the geometrical relationship between UAV images is often complex due to variations in ground relief, changes in imaging viewpoints, and shooting at low altitudes, where image pairs cannot be accurately matched using a parametric transformation model (e.g., affine or homography) as in most existing methods. Image matching is a critical prerequisite of image mosaic, which aims to overlay two images of the same scene geometrically [10,11,12,13]. On the basis of the type of data given, image matching can be divided into rigid and nonrigid. In traditional mosaic matching, the implicit constraint is that the given image is rigid. However, for UAV images, nonrigid matching is crucial because these images often contain local deformation, which cannot be solved by rigid matching. One the other hand, remote sensing images captured by UAVs are from a low altitude, and the resulting images cannot be approximated as a plane. In the traditional stitching system, single planar perspective transform (homography) [14] is used with bundle adjustment for optimization, and this usage leads to ghosting errors. To overcome such limitation, Chin et al. proposed as-projective-as-possible (APAP) image stitching [15], which meshes images, and each grid is aligned with a homography, which is suitable for UAV image mosaic. However, the method has a prerequisite that the images have been accurately aligned.

To address the aforementioned challenges, we introduce a nonrigid matching algorithm on the basis of motion field interpolation, namely, vector field consensus (VFC) [16], to the mosaic system for generating accurate feature matching on remote sensing images. In bundle adjustment [17], we propose a new strategy that improves the original relationship of homography conversion and use the homography response to perform bundle adjustment, which is robust to our image mosaic. The experiments on different sets of UAV images reveal the superiority of our method.

## 2. Related Work

This section briefly reviews the background material on which our work is based, including nonrigid matching algorithms, local transformation descriptions, bundle adjustment, and aerial image mosaic.

### 2.1. Nonrigid Image Matching

The image matching phase is a key step in image stitching and many other computer vision tasks [18,19,20,21]. In the matching phase, the capability to minimize registration errors plays an important role in the subsequent steps of the system. Here, the matching problem aims to align the overlapping regions of two images pixel by pixel. Obtaining a good stitching effect requires a dense pixel-by-pixel matching of the overlapping regions. When the unknown complex relationship between images cannot be accurately modeled by a specific model, especially when a nonrigid change occurs in the image, achieving accurate pixel-by-pixel registration is difficult.

In general, the SIFT [22] feature points are extracted to register the relationship between images. Sparse feature point matching is used to guide dense pixel-by-pixel registration. Coping with a large proportion of feature points is possible with a general linear model, such as RANSAC [23], and its variants, including MLESAC [24], LO-RANSAC [25], and PROSAC [26], to estimate the spatial transformation between images. This approach seems to be a good solution. However, this series of algorithms relies on specific geometric parameter models. Images with nonrigid changes can no longer be applied because the transformation cannot be modeled parametrically.

Li and Hu [27] proposed an identifying correspondence function (ICF) algorithm based on nonparametric models to achieve nonrigid matching, but its matching precision drops sharply when many outliers are present Another strategy for solving the matching problem is to estimate the corresponding matrix of the two point sets in combination with the parameter or nonparametric geometric constraints. The method based on estimating the corresponding matrix jointly estimates the correspondence between the point sets and transformation, compared with the previous independent estimated point correspondence and transformation. Representative methods for finding inter-image correspondence matrices, such as iterated closest point (ICP) [28], coherence point drift (CPD) [29] and its variant [30], and locality preserving matching (LPM) [31,32], have also been proposed to match nonrigid cases, but such methods generally cannot tolerate excessive outliers. Moreover, for robustness purposes, such algorithms typically impose penalties on points that are not matched. The point set matching problem can also be solved by solving a graph matching problem [33]. Such methods construct an affinity matrix between point sets and then perform spectral analysis to obtain ordered features of the point set. These methods include dual decomposition (DD) [34], spectral matching (SM) [35], and graph shift (GS) [36]. However, the computational complexity of these methods is typically high, which cannot be applied to address large-scale real-time matching tasks. The VFC algorithm proposed by Ma et al. is also based on the nonparametric model [16]. This algorithm converts the point set matching problem into vector field interpolation and can efficiently generalize sparse matching to dense matching, which is suitable for our UAV image matching.

### 2.2. Local Transformation

The conversion relationship of the overlapping image alignment is estimated after acquiring the image matching relationship. Then, the aligned images are combined into a common plane. In the traditional stitching method, a homography is used to represent this conversion relationship. The goal is to minimize alignment errors between overlapping pixels by uniform variations. A homography is a commonly used transformation because it maintains the most flexible transformation of all lines, and the resulting panorama does not produce considerable distortion. The homography has excellent effect for all images in a plane. However, for the UAV images tackled in this study, this ideal condition typically does not hold.

As a homography cannot efficiently align the pixels of the overlapping part, Liu et al. [37] proposed a content-preserving warp that minimizes the registration error and maintains the stiffness of the scene. However, in image stitching, large differences are observed in the rotation and translation between different views. The interpolation of this method is not flexible enough due to the constraint of stiffness. Gao et al. [38] divided a scene into distant and ground planes that sweep out from the camera’s location; they aligned the two planes with two homographies and then blended these to align the images. This method is more flexible than using a single homography, but it remains unsuitable for complex scenarios. Lin et al. [39] used a smoothly varying affine warp to align the image. The local deformability and alignment capability were strong and the method can handle parallax. However, the use of affine regularization is appropriate for interpolation, which may not be optimal for extrapolation, and the affinity is insufficient for completing the perspective transformation. Chin et al. [15] proposed a transformation that is as projected as possible by using an algorithm that is as transformable as possible. The purpose of the deformation is to perform a global perspective transformation and allow local non-projection transformations by sampling the local transformation model for image mosaic. The local model has high degree of freedom and is flexible, which can handle local transformation and reduce ghosting problems. Subsequent methods use global similarity to mitigate projection distortion in nonoverlapping regions, such as shape-preserving half-projective [40]. This approach adds constraints, combined with homography and similarity transformations, corrects the shape of the stitched image, and reduces projection distortion. Adaptive as natural as possible [41] also uses global similarity transformations to correct shapes, but it can adaptively determine angles to correct image shapes efficiently.

### 2.3. Bundle Adjustment

Given a set of overlapping images, we aim to project all images onto a common surface. This process will inevitably accumulate errors. To obtain an improved stitching effect, we must simultaneously refine these errors. The current method [4,42] optimizes the focal length of all views and the relative rotation of the camera pose through bundle adjustment and then aligns the series of images. The bundle adjustment can be based on the projection of all points in the image while extracting the 3D point coordinates describing the scene structure, the relative motion parameters, and the optical parameters of the camera.

Bundle adjustment is often used in feature-based 3D scene reconstruction algorithms as the final step. This method is based on 3D structure and perspective parameters (e.g., camera position, orientation, inherent calibration, and radial distortion). Providing an initial estimate, the bundle adjustment simultaneously refines motion and structural parameters by minimizing projection errors between observed and predicted image points. The best reconstruction effect is obtained under the assumption that the obtained image features contain noises.

The ultimate goal of the bundle adjustment is to reduce the positional projection transformation (re-projection) error between the points of the observed and reference images (predicted image) by using a least squares algorithm. The most successful strategy is Levenberg-Marquardt algorithm [43], which is easy to implement and can quickly converge on a wide range of initial estimates.

### 2.4. Aerial Image Mosaic

The theme of this study is the stitching of UAV images. In this part, we introduce the related work in aerial image mosaic.

Park et al. [44] proposed an algorithm for hierarchical multi-level image mosaicing for the autonomous navigation of UAV. The algorithm prevents the error accumulation propagated along the frames by incrementally building a long-duration mosaic on the fly, which is hierarchically composed of short-duration mosaics. The real-time processing requirements in autonomous navigation are fulfilled. In particular, the system can automatically adapt to the scene changes of the images to be spliced and can automatically select suitable methods for feature selection. This point is important in the autonomous navigation of UAV. The proposed system is a causal system and is suitable for real-time application. Ghosh et al. [45] proposed a spatial domain-based super-resolution mosaicing system and presented its evaluation results. This algorithm combines image mosaicing and super-resolution-reconstruction algorithms and has the characteristics of robustness and simple calculation. The authors used spatial domain-based super resolution and made it practical in real-time applications, such as surveillance and remote sensing. Ghosh et al. also introduced indicators for the quantitative evaluation of image mosaicing in multiple scene categories [46], which contributed to the defects in this aspect.

## 3. Methodology

Our task is to place a set of UAV images together in a system to obtain a mosaic panorama. The previous section indicates that the traditional stitching system cannot produce good results because of the particularities of UAV images. Hence, we perform modifications to the system to make it robust to UAV images.

### 3.1. Nonrigid Feature Matching Using Vector Field Consensus

In the feature matching stage, the SIFT descriptor is used to conduct initial feature matching and obtain the corresponding relationship of feature points between images. UAV images may have nonrigid changes. Thus, we introduce a nonrigid matching algorithm based on VFC [16].

Given a set of putative SIFT matches $S={\left\{\left(x_{n},y_{n}\right)\right\}}_{n=1}^{N}$, our purpose is to distinguish false matches (outliers) from the correct matches (inliers) and estimate the transformation f to fit the inliers efficiently, where f∈H. We assume that f is nonrigid and H is a reproducing kernel Hilbert space (RKHS) [47].

We assume that the noise of inliers is Gaussian with zero mean and uniform standard deviation σ. The outliers are uniformly distributed in the image domain, and the distribution is assumed to be uniform 1/a, where *a* is a constant. Let γ be the percentage of inliers that we do not know in advance. Thus, the likelihood is a mixture model of distributions for inliers and outliers:(1)$$\begin{array}{cc} p(Y|X,\theta ) & =\prod_{n=1}^{N}p\left(y_{n},z_{n}|x_{n},\theta \right)\\ & =\prod_{n=1}^{N}\left(\frac{\gamma }{{\left(2\pi \sigma^{2}\right)}^{D/2}}e^{-\frac{∥y_{n}-f\left(x_{n}\right){∥}^{2}}{2\sigma^{2}}}+\frac{1-\gamma }{a}\right), \end{array}$$
where $\theta =\{f,\sigma^{2},\gamma \}$ is the set of unknown parameters. We associate sample *n* with a latent variable $z_{n}\in \left\{0,1\right\}$, where $z_{n}=1$ represents Gaussian distribution and $z_{n}=0$ represents uniform distribution.

Considering the smoothness constraint, the prior of f can be written as
(2)$$p\left(f\right)\propto e^{-\frac{\lambda }{2}{∥f∥}_{H}^{2}}.$$

By applying Bayes rule, p(θ|X,Y)∝p(Y|X,θ)P(f), we estimate θ as an MAP solution, i.e., $\theta^{*}$:(3)$$\theta^{*}=\arg\max_{\theta }p\left(Y|X,\theta \right)p\left(f\right),$$
where $\theta^{*}$ corresponds to the estimate of the true θ. Thus, the transformation f can be obtained. We introduced and solved a latent variable $z_{n}$ by using the EM algorithm. We can write the complete-date log posterior as follows [48]:(4)$$\begin{array}{c} Q\left(\theta ,\theta^{\mathrm{old}}\right)=-\frac{1}{2\sigma^{2}}\sum_{n=1}^{N}p_{n}{∥y_{n}-f\left(x_{n}\right)∥}^{2}+\ln\gamma \sum_{n=1}^{N}p_{n}\\ +\ln\left(1-\gamma \right)\sum_{n=1}^{N}\left(1-p_{n}\right)-\frac{D}{2}\ln\sigma^{2}\sum_{n=1}^{N}p_{n}-\frac{\lambda }{2}{∥f∥}^{2}, \end{array}$$
where $p_{n}=P\left(z_{n}=1|x_{n},y_{n},\theta^{\mathrm{old}}\right)$. The value may be maximized by regarding $z_{n}$ as missing data from the mixture model.

**E-step**: Denote $P=\mathrm{diag}(p_{1},\cdots ,p_{N})$, where the responsibility $p_{n}$ can be computed by applying the Bayes rule:(5)$$p_{n}=\frac{\gamma e^{-\frac{∥y_{n}-f\left(x_{n}\right){∥}^{2}}{2\sigma^{2}}}}{\gamma e^{-\frac{∥y_{n}-f\left(x_{n}\right){∥}^{2}}{2\sigma^{2}}}+{\left(2\pi \sigma^{2}\right)}^{\frac{D}{2}}\left(1-\gamma \right)/a},$$

The posterior probability $p_{n}$ is a soft decision, which indicates that the degree of sample *n* agrees with the current estimated f.

**M-step**: We determine the revised parameter estimate $\theta^{\mathrm{new}}$ as follows: $\theta^{\mathrm{new}}={\arg\max}_{\theta }Q\left(\theta ,\theta^{\mathrm{old}}\right)$. By using the derivative of Q(θ) with respect to variance $\sigma^{2}$ and the mixing coefficient γ and setting them to zero, we obtain
(6)$$\sigma^{2}=\frac{\sum_{n=1}^{N}p_{n}{∥y_{n}-f\left(x_{n}\right)∥}^{2}}{D\sum_{n=1}^{N}p_{n}},$$
(7)$$\gamma =\frac{\sum_{n=1}^{N}p_{n}}{N}.$$

To complete the EM algorithm, the vector field f should be estimated in the M-step, which can be obtained by solving a regularization problem [16].

### 3.2. Local Wrap and Bundle Adjustment

After matching two images, the source image can be projected onto the target image through homography. However, the errors accumulate and magnify when stitching multiple images, especially in multiple overlapping regions [49]. We can simultaneously optimize projection functions to register panoramic images under the framework of minimum mean square error and reduce the cumulative error.

The first step when stitching multiple images is to find the reference plane [50,51] to which all images are projected through a basic homographic warp. We select an image from the input image as the reference plane, as described in [4]. All overlapped images are identified through the feature matching of the panoramic image. We select the image with the most overlapped parts as the reference plane so that other images can be projected to the reference plane by a homographic warp. The traditional method uses direct linear transformation (DLT) [52] to calculate the single response between images (which requires four pairs of matched points). The registration of UAV images cannot be changed simply by a single response. Inspired by Chin et al. [15], the registration error caused by a single response was effectively reduced by differentiating the image, and each differential corresponds to a basic homographic matrix.

Given two overlapping images *I* and $I^{\prime }$ and their matched points $x_{i}={\left(x_{i},y_{i}\right)}^{T}$, $y_{i}={\left(x_{i}^{\prime },y_{i}^{\prime }\right)}^{T}$, i=1,…,n, the global transformation between two images can be estimated by
(8)$$\tilde{y}∼H\tilde{x},$$
where $\tilde{x}$ denotes x in homogeneous coordinates, ∼ indicates the equality up to scale, and H is the global homography that is a 3×3 matrix.

In 2D projective warp, DLT is a basic method for estimating **H** from a set of noisy point matches. We vectorize H into a vector h. Let $a_{i}$ be the two rows of the matrix for two matched points. Given an estimate h, the quantity $∥a_{i}h∥$ is the algebraic error. DLT minimizes the sum of squared algebraic errors:(9)$$\hat{h}=\arg\min_{h}\sum_{i=1}^{N}{∥a_{i}h∥}^{2}.$$

By stacking vertically $a_{i}$ for all *i* into matrix A of size 2N×9, the problem can be rewritten as
(10)$$\hat{h}=\arg\min_{h}\sum_{i=1}^{N}{∥Ah∥}^{2}.$$

The solution is the least significant right singular vector of A. Given the estimated H (reconstructed from $\hat{h}$), in the 2D projective warp, an arbitrary pixel $x_{*}$ in the source image *I* is warped to the position $y_{*}$ in the target image $I^{\prime }$ by
(11)$${\tilde{y}}_{*}∼H{\tilde{x}}_{*},$$

For UAV images, global homography cannot efficiently match two images. In this case, we introduce a location-dependent homography.
(12)$${\tilde{y}}_{*}∼H_{*}{\tilde{x}}_{*}.$$

Each pixel $x_{*}$ corresponds to a location-dependent homography $H_{*}$, where $x_{*}$ is estimated from the weighted problem
(13)$$h_{*}=\arg\min_{h}\sum_{i=1}^{N}{∥w_{*}^{i}a_{i}h∥}^{2}.$$

We can define the scalar weights ${\left\{w_{*}^{i}\right\}}_{i=1}^{N}$ as
(14)$$w_{*}^{i}=\exp(-∥x_{*}-x_{i}{∥}^{2}/\delta^{2}),$$
where δ is a scale parameter. The exact solution steps are the same as before.

When we have a set of location-dependent homographies ${\left\{H_{*}^{k}\right\}}_{k=1}^{K}$, where each $H_{*}^{k}$ corresponds to a pair of matching points in two images, we should consider the cumulative error during image mosaic. We minimize the cost to minimize the transfer error of all correspondences:(15)$$E_{*}\left(\Theta \right)=\sum_{i=1}^{N}\frac{w_{*}^{i}}{\sum_{k=1}^{K}\xi_{ik}}\sum_{k=1}^{K}\xi_{ik}{∥x_{i}^{k}-f\left(p_{i},H_{*}^{k}\right)∥}^{2},$$
where $\Theta =[H_{1}^{k},\ldots ,H_{*}^{k},p_{1},\ldots ,p_{N}]$ and f(p,H) is the project warp defined as
(16)$$f\left(p,H\right)={\left[\frac{r_{1}{\left[p^{T}1\right]}^{T}}{r_{3}{\left[p^{T}1\right]}^{T}}\;\frac{r_{2}{\left[p^{T}1\right]}^{T}}{r_{3}{\left[p^{T}1\right]}^{T}}\right]}^{T},$$
where $r_{1}$, $r_{2}$ and $r_{3}$ are the three rows of homography **H**. We introduce a parameter $\xi_{ik}\in \left\{0,1\right\}$, where $\xi_{ik}=1$ indicates that correspondence exists; otherwise, $\xi_{ik}=0$.

## 4. Experimental Results

We compare our method with the traditional image mosaic method. In the experiment, the UAV images have nonrigid changes, which cannot be treated as a plane. In the following, we initially report the feature matching results and then provide the image mosaic results.

### 4.1. Remove Mismatches on Nonrigid Image Pairs

In stitching remote sensing images, a nonrigid matching algorithm must be added because of the high mismatching rate of remote sensing images. The false matching points of UAV images can be removed effectively by adding the VFC algorithm. To evaluate the feature matching performance of VFC, we use three other widely used methods for comparison, namely, RANSAC [23], ICF [27], and GS [36]. We select two representational image pairs [32] that have local nonrigid distortion for evaluation, where the ground truth feature matches are supplied by the dataset. Two common evaluation indicators, namely, precision and recall, are used to measure the performance of our algorithm [53]. The precision rate is defined as the ratio of the number of retained inlier points to the total number of initial feature points, and the recall rate is defined as the ratio of the number of retained inlier points to the number of initial inlier points. Experimental results show that this method is better than other methods when performing nonrigid registration. Figure 1 and Table 1 shows the results. VFC can achieve better matching performance on the UAV images than other state-of-the-art methods.

### 4.2. Overall Differentiation and Local Wrap

In projection transformations, the traditional method (Autostitch) [4] completes the transformation by using a homography matrix. Parallax changes in the image will bring negative effects. To solve these problems, we adopt the idea of differentiation. This method can tolerate the parallax in the image. We present a simple example in Figure 2 to illustrate the role of this idea in projection transformation. Local and global warps will have different effects in the stitching, especially when the two images are not on one plane (the two views differ by a rotation and translation). In Figure 2, a homography matrix can efficiently retain the rigid change of the image, but it is bound to leave matching points. Instead, our differentiation method can include most (even all) of the match points. Although the image stiffness may be sacrificed, the effect is acceptable. We also perform the experiment, and Figure 3 shows the results. In Figure 3, strictly aligning these images through the traditional method is difficult. Ghosts in the mosaic image are evident, because a homography matrix fails to match the objects in the two images efficiently.

Our method can greatly improve the situation. The image in the third row of Figure 3 is our mosaic result with evidently few ghosts. White boxes are used to highlight the ghostly parts. Moreover, we compare the difference in the effects of the two strategies. Our method is improved with local homography.

Differentiation is robust to local nonrigid transformation of images and can solve the problem that an image is not on a plane. In this way, the problem of local projection mismatch can be reduced (reducing the ghosting errors in image mosaic), and the performance of the subsequent bundle adjustment can be improved. We also conduct a series of experiments to verify the effect of our method. We compare our method not only with the traditional method (Autostitch) [4] but also with other state-of-the-art methods, including AANAP [41], SPHP [40], and parallax tolerant (PT) [54].

The results are reported in Figure 4, Figure 5 and Figure 6. Red boxes highlight the obvious parts of the ghost effect in the mosaic results. We enlarge the area in red boxes to make results clearer. Results show that our method is robust for stitching UAV images and can get better results than the competitors. The mosaic of panoramic image is based on the mosaic of two images. We can achieve improved results here, which alleviate the pressure of subsequent panorama stitching. However, our results still have shortcomings, and the ghosting of some parts cannot be completely eliminated, which can be further improved during bundle adjustment.

### 4.3. The Effect of Nonrigid Changes and Parallax

In this section, we will analyze how the mentioned challenges (such as nonrigid changes caused by ground relief, and parallax caused by changes in imaging viewpoints) affect the mosaic result.

Some experiments show the effect of those challenges to the mosaicing results. In Figure 4, nonrigid changes are observed in the ground. In the registration stage, the traditional method cannot efficiently align these nonrigid changes and it is easy to form many mismatches. The large number of small stones in the scene has formed a ghost because they are not efficiently matched. We enlarged the scene in the red boxes. Results are displayed next to the mosaicing results. Our method can improve the ghost effect. Figure 5 and Figure 6 show the images obtained in two different scenes, but they have a certain parallax. In Autostitch, the overlapping areas will be misaligned when the parallax image is projected onto a common plane. There are some unsatisfactory areas in the mosaicing result. In Figure 5, the ground lines in the red boxes are clearly wrong in Autostitch. Such a mosaic result is not what we want to see. Other advanced methods can almost alleviate these ghosts. Our methods can acquire better results than these methods. Our results are clear and the line is still correct. In Figure 6, there are duplicate parts about cars and roads in the red boxes. The stitching results in the last row are better than the rest images. Our method is more suitable for such parallax scene.

At the same time, we choose some sequential images without parallax and nonrigid changes to carry out the experiment. Experimental results are shown in Figure 7. We can see that the results of Autostitch are almost identical to the results of our approach. In the Figure 8, these source images have some parallax and nonrigid changes. The result of Autostitch has errors, where the challenges mentioned above lead to these errors.

### 4.4. The Effect of Noise and Blurriness

When we acquire UAV images, introducing noise and blur into the images is inevitable. Therefore, a good system must resist the intrusion of noise and blur. We also conduct experiments on this aspect. Each of these images are convolved with a Gaussian blurring kernel of SD=1.2 (standard deviation). The images are also contaminated by additive white Gaussian noise with zero mean and variance 0.0004. We use these two strategies to evaluate the behavior of our mosaicing algorithm in the presence of blur or noise.

Figure 9 shows the results. Noise and blur in general have no effect on our system, because the ghost in the mosaic image is not enhanced or enlarged. Thus, the algorithm can still work efficiently and remove mismatched feature points further accurately in the stage of removing mismatches. In the registration stage, the image is unaffected by noise and blur. The algorithm can precisely align the pixels and then project the image to a common plane.

### 4.5. Bundle Adjustment and Panoramic Stitching

At this stage, our method has the same capability as Autostitch and can achieve the mosaic in a group of arbitrary images. Hence, the source image can be any image of a scene. In Autostitch, only one global homography is used for stitching. Thus, the bundle adjustment optimizes the relative rotation and homographies between sets of overlapping images. This case only requires average intensity blending of the aligned image. Results show that our stitching system has a better effect than the traditional stitching system, and the ghosting effect is considerably reduced. Figure 8 presents the results.

As shown in Figure 3 and Figure 8, when the traditional method (Autostitch) [4] mosaics the panorama image, the error is cumulatively enlarged. When Autostitch mosaics two images, the ghosting effect is not evident. When images are added to the system, the bundle adjustment cannot efficiently remove the accumulated error. By contrast, our method can efficiently remove the accumulated error during bundle adjustment. In Figure 10 and Figure 11 [55], there are some images of arbitrary viewpoints and we also compare the splicing results. In Figure 10, errors are displayed when Autostitch stitches source images. This indicates that the method cannot process these images, so we do not report the result of Autostitch, and only give the splicing result of our method. In Figure 11, The Autostitch stitching results still have some misaligned areas. We have highlighted it in red boxes and zoomed in on the side.

Our method sacrifices the stiffness of the stitching image. The effect is good when stitching two images. However, distortion is observed in the result when stitching multiple images. This situation can be seen from our stitching results in Figure 7, Figure 8, Figure 9, Figure 10 and Figure 11, but the overall effect is acceptable. For most UAV images, the mosaic results obtained by our system are acceptable. The distortion will affect the image effect only when the parallax of the source image is particularly large. This case can be improved later, i.e., how to obtain a good mosaic effect on a large-parallax image.

## 5. Conclusions

In this paper, we propose an image mosaic method based on robust feature matching and a new bundle adjustment strategy. The use of robust feature matching produces further accurate results and reduces the pressure on the subsequent steps of the system. When calculating the transformation relation between images, we differentiate the global homography matrix so that the new adjustment must only blend the aligned image with the average intensity. Results show that our approach provides a further natural panorama with no visible parallax in the overlapping regions and that the ghost effect produced by mosaicing is evidently reduced. Our system is more suitable for UAV images and has a wider application range than the traditional image mosaic method. Future research developments will improve the image effect in the case of a large parallax, which can be performed by integrating seam-cut methods into stitching framework.

## Figures and Tables

**Figure 1 sensors-19-01898-f001:**
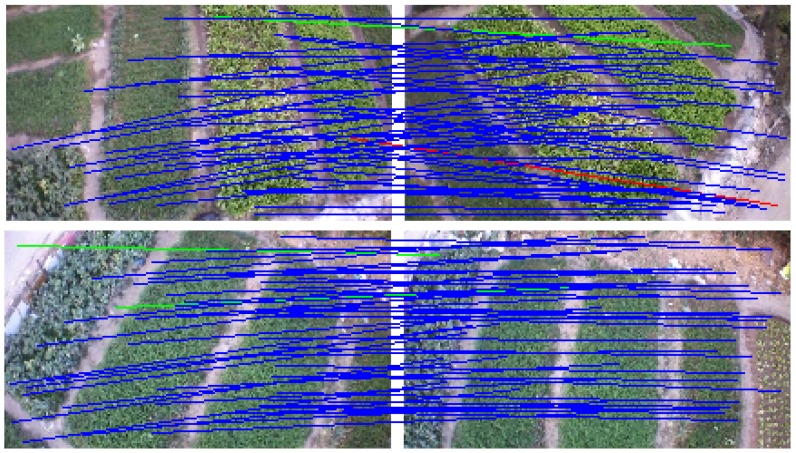
Experimental results on two typical image pairs of nonrigid objects. The lines indicate mismatch removal results (blue = true positive, green = false negative, red = false positive). For visibility, in the image pairs, only 50 randomly selected correspondences are presented, and the true negatives are not shown.

**Figure 2 sensors-19-01898-f002:**
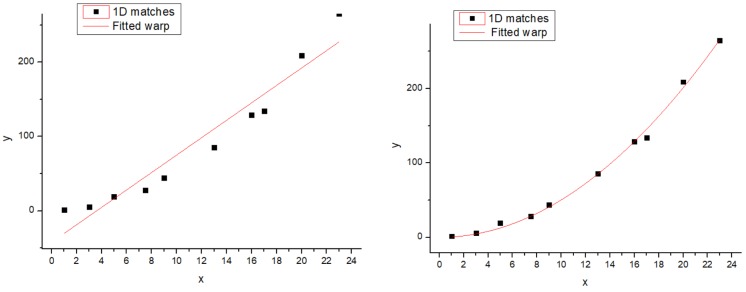
To generate a 1D analogy of image stitching, a set of 1D matches ${\left\{x_{n},y_{n}\right\}}_{n=1}^{N}$ are generated by projecting a 2D point cloud onto two 1D image “planes”. Here, the two views differ by a rotation and translation. The first indicates that the match points are warped by a global homography and it is unable to model the local deviations of the data. The second indicates that the match points are warped by some local homography and it interpolates the local deviations flexibly.

**Figure 3 sensors-19-01898-f003:**
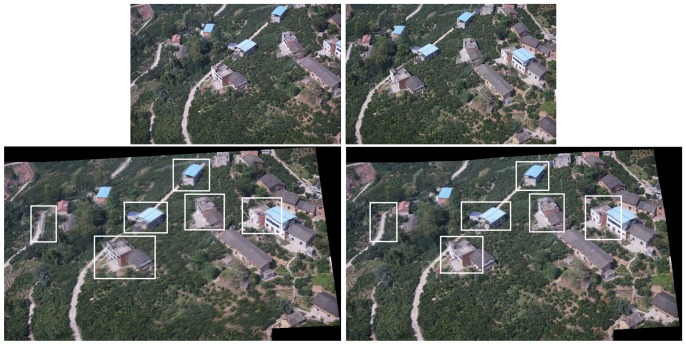
Mosaic results. The first row is the two images to be stitched together. The second row is two stitched images by global homography and local homography, respectively. White boxes highlight the ghostly parts.

**Figure 4 sensors-19-01898-f004:**
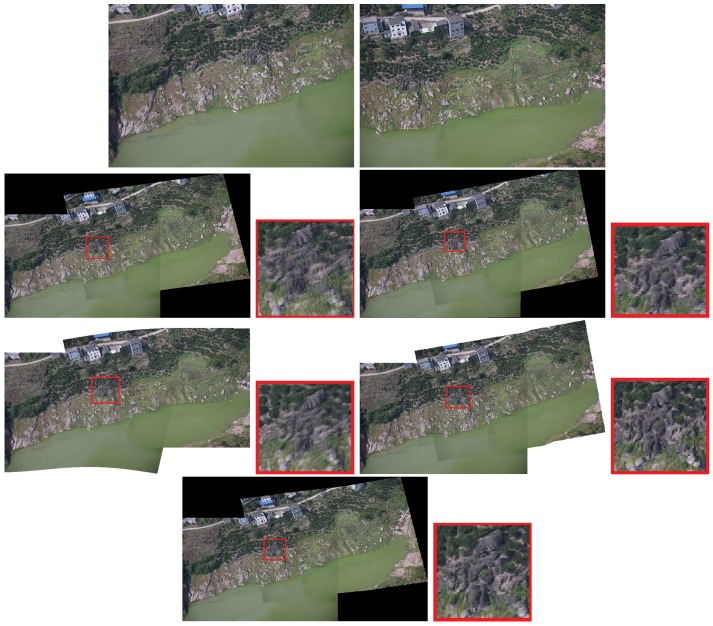
Qualitative comparison on UAV image mosaic. The first row is the two images to be stitched. The second and third rows are the stitching results of Autostitch [4], AANAP [41], SPHP [40], and PT [54]. Our result is shown in the last row. Red boxes highlight the ghostly parts.

**Figure 5 sensors-19-01898-f005:**
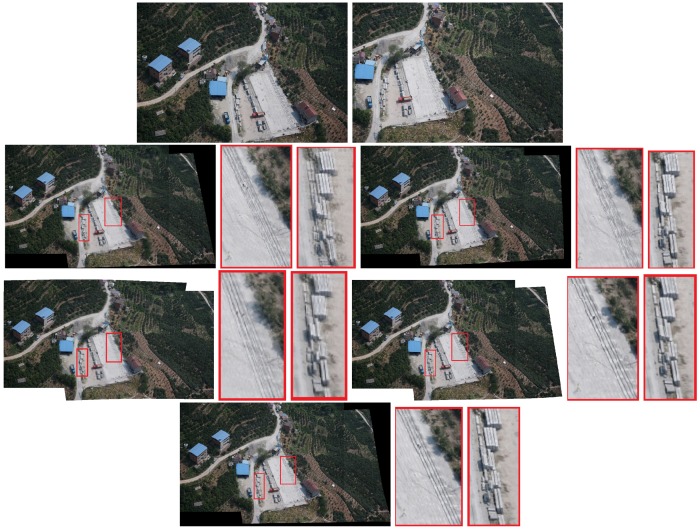
Qualitative comparison on UAV image mosaic. The first row is the two images to be stitched. The second and third rows are the stitching results of Autostitch [4], AANAP [41], SPHP [40], and PT [54]. Our result is shown in the last row. Red boxes highlight the ghostly parts.

**Figure 6 sensors-19-01898-f006:**
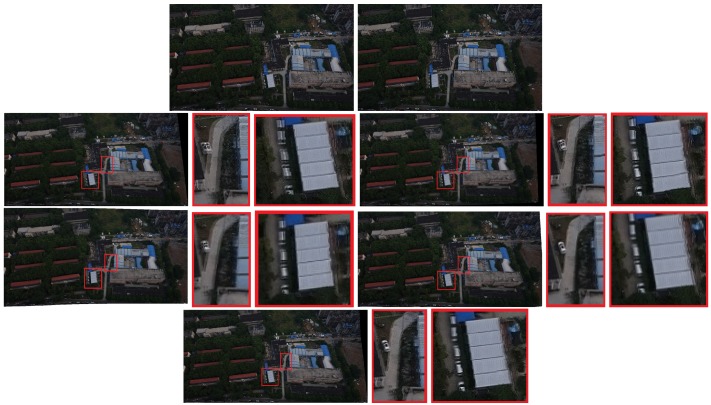
Qualitative comparison on UAV image mosaic. The first row is the two images to be stitched. The second and third rows are the stitching results of Autostitch [4], AANAP [41], SPHP [40], and PT [54]. Our result is shown in the last row. Red boxes highlight the ghostly parts.

**Figure 7 sensors-19-01898-f007:**
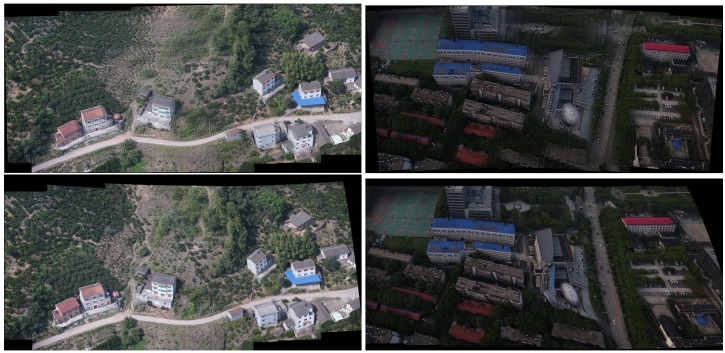
Experimental results of panoramic stitching. The source images are sequential images captured from left to right without nonrigid changes and parallax. The first row is the results of Autostitch. The second row is the results of our method.

**Figure 8 sensors-19-01898-f008:**
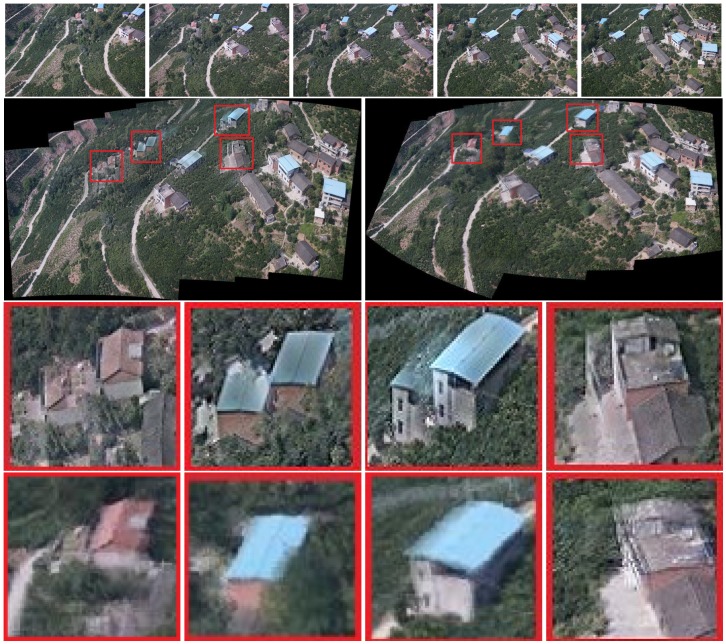
Experimental results of panoramic stitching. The first row is the image to be stitched together. The second row is the results of Autostitch and our method. Red boxes highlight the ghostly parts. The last two rows are the details of the red boxes.

**Figure 9 sensors-19-01898-f009:**
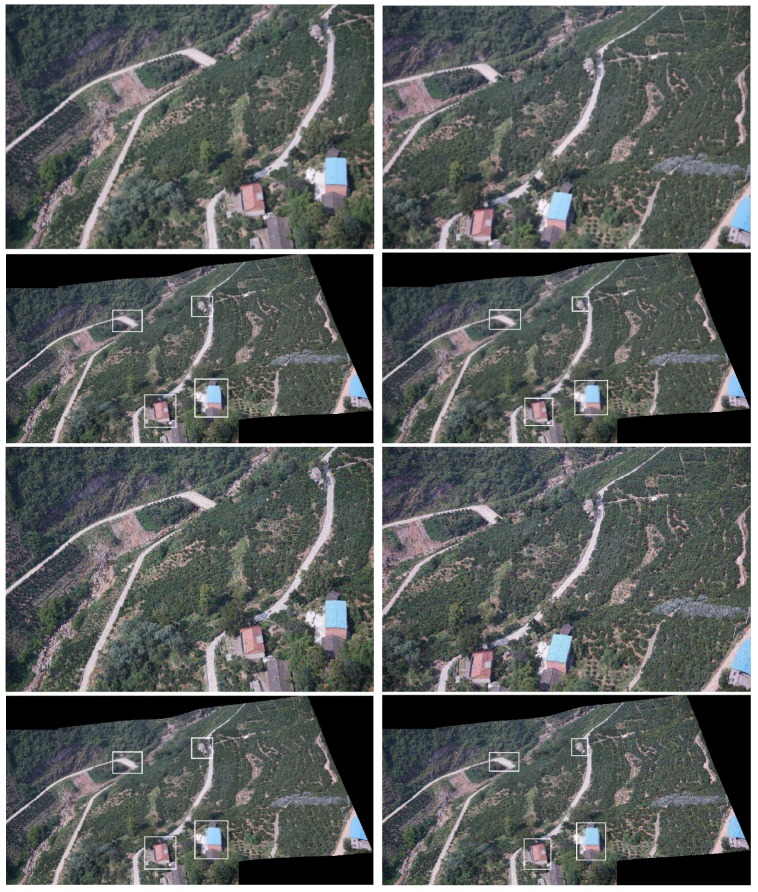
The first line is the blurred source image. The second line is the mosaic result of the unblurred source image and the mosaic result of the blurred source image. The third line is the noisy source image. The fourth line is the mosaic result of the source image without noise and the mosaic result of the source image with noise. White boxes highlight the ghostly parts.

**Figure 10 sensors-19-01898-f010:**
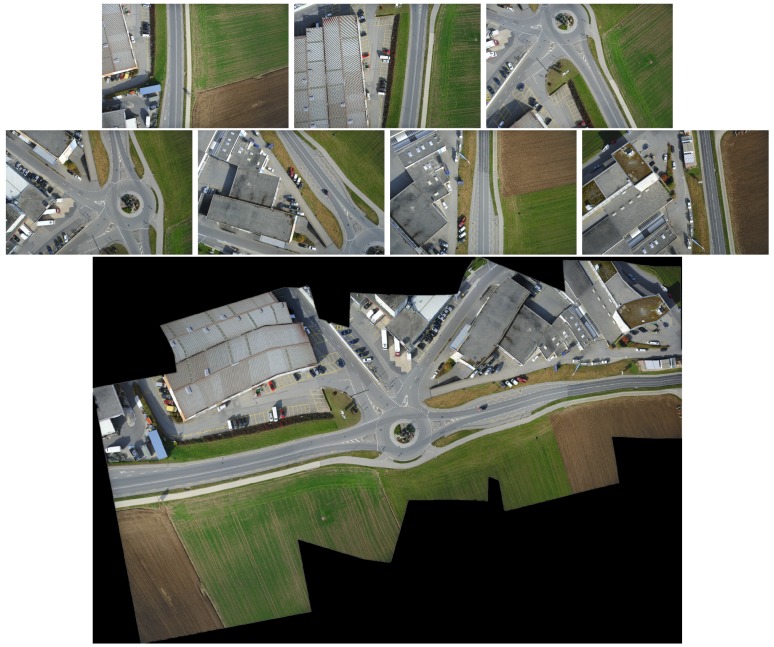
Experimental results of panoramic stitching. The first and second rows are the images to be stitched together. The third row is the result of our method. Autostitch cannot process these images.

**Figure 11 sensors-19-01898-f011:**
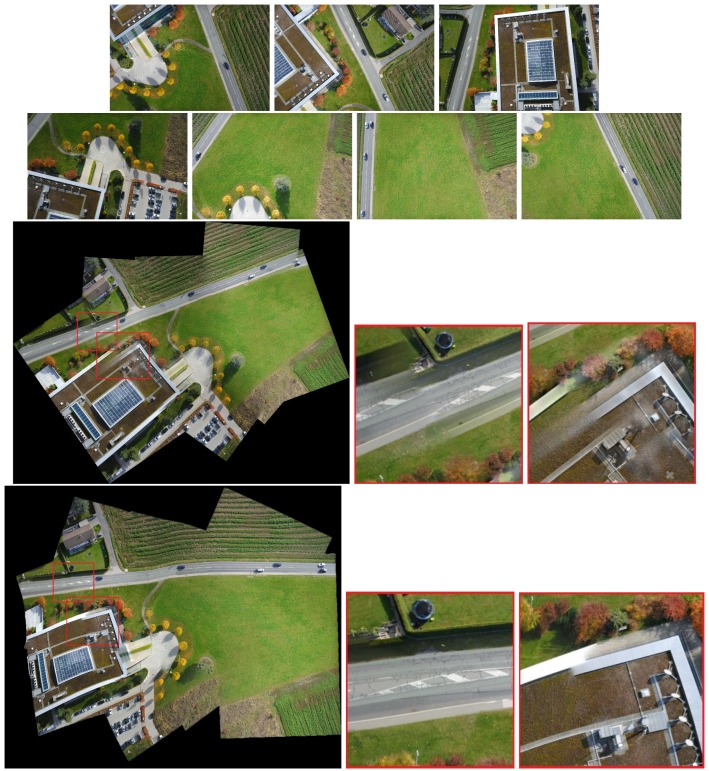
Experimental results on panoramic stitching. The first and second rows are the images to be stitched together. The third and fourth rows are the results of Autostitch and our method. Red boxes highlight the ghostly parts.

**Table 1 sensors-19-01898-t001:** The precision (P) and recall (R) results of different methods on the image pairs in Figure 1. The initial inlier percentage is 37.27% and 42.39% on the two image pairs, respectively.

	RANSAC [23]	ICF [27]	GS [36]	VFC
(P, R)	(99.66%, 97.67%)	(98.98%, 91.76%)	(99.59%, 81.51%)	(99.67%, 98.02%)
(P, R)	(100.0%, 97.07%)	(99.32%, 92.86%)	(100.0%, 94.06%)	(100.0%, 97.62%)

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
