# Peer review of "A Robust Method for Automatic Panoramic UAV Image Mosaic"

_sensors, 2019, doi:10.3390/s19081898_

Round 1
Reviewer 1 Report
This paper proposed an automatic image mosaic method for unmanned aerial vehicle (UAV) images. A non-rigid matching scheme (based on vector field consensus) has been employed to obtain accurate feature matching between each pair of adjacent (overlapping) images.
A new bundle adjustment strategy has been proposed to achieve the image mosaic task.
To verify the method, Figure 3~6 demonstrate the resulting mosaic images. However, these images were captured at the same location and thus they all show very similar scenery. There is lack of experimental variety to support the proposed approach. Moreover, there is no information about the encountered challenges of the chosen set of images (e.g., the ground relief and changes in imaging viewpoints). It is suggested to conduct a set of synthetic experiments to analyze how the mentioned challenges affect the mosaicing result.
In the experiments, the source images were captured either from left to right or right to left, but in real-world situation, sometime it is necessary to stitch a set of images captured at arbitrary viewpoints.
Author Response
We would like to thank the reviewer for the constructive comments, which enable us to greatly improve the quality of our manuscript. We have provided a marked-up copy to highlight the changes in our revision. Next, we provide point-by-point responses to the review comments.
1) To verify the method, Figures 3-6 demonstrate the resulting mosaic images. However, these images were captured at the same location and thus they all show very similar scenery. There is lack of experimental variety to support the proposed approach.
Reply: Thanks for this suggestion. In our revision, we have added several image sets of different types of scene for stitching, as shown in Figures 3-8. In these different types of scene, our method can consistently produce satisfying results. These experimental results make our method more convincing.
2) Moreover, there is no information about the encountered challenges of the chosen set of images (e.g., the ground relief and changes in imaging viewpoints). It is suggested to conduct a set of synthetic experiments to analyze how the mentioned challenges affect the mosaicing result.
Reply: Thanks for this suggestion. As for this problem, we did not describe it in detail in our previous manuscript. All the above mentioned challenges will bring ghost effects to the stitching results, and our method is mainly to eliminate these ghost effects. We have described these challenges in detail and added some experimental results (i.e., Figs. 4-6) in the revised paper. Please refer to the last paragraph in Sec. 4.2 on Page 9 as follows.
“The results are reported in Fig. 4, Fig. 5 and Fig. 6. In Fig. 4, there are some non-rigid changes in the ground, and it is easy to form many mismatches in traditional methods. The large number of small stones in the scene have formed a ghost because they are not well matched. Our approach is more robust for this situation. In Fig. 5 and Fig. 6, they are the images obtained in two different scenes, but these images all have a certain parallax. As can be seen from the corresponding results, our method is more robust for stitching UAV images. The ghost effect is very obvious in Autostitch. Such a mosaic result is not what we want to see. Other advanced methods can more or less alleviate these ghosts. Our methods can get better results than these methods. The mosaic of panoramic image is based on the mosaic of two images. We can get better results here, which alleviates the pressure of subsequent panorama stitching. However, our results still have some shortcomings, and the ghosting of some parts cannot be completely eliminated, which can be further improved during the bundle adjustment phase.”
3) In the experiments, the source images were captured either from left to right or right to left, but in real-world situation, sometime it is necessary to stitch a set of images captured at arbitrary viewpoints.
Reply: Thanks for pointing out this problem. In the bundle adjustment phase, our method has the same ability as the Autostitch, which can achieve the mosaic in a group of arbitrary images. This means that the source images can be any images of a scene. The difference is that Autostitch cannot process source images with parallax, and our method can handle this situation. We have provided an explanation of this point in the section of experimental results of the revised paper. Please refer to the beginning of Sec. 4.4 on Page 11.
Reviewer 2 Report
In this paper, a non-rigid matching algorithm is proposed to generate accurate feature matching on remote sensing images. We also propose a new strategy for bundle adjustment to make the mosaic system suitable for UAV image panoramic mosaic effect.
1) The paper needs some English editing
· Line 1: Study should be replaced by Paper
· Line 2: This second sentence should be revised
· Line 16: What does it mean “good safety”?
· …..
2) The authors state in the abstract that their method outperforms traditional methods. The Experimental Results they state that their method is compared to the traditional image mosaic method. They are inconsistencies in the authors’ statements. I also do not see any comparison in the paper
3) I do not see any quantitative evaluation in the paper. The authors can refer to the following about metrics to evaluate their algorithm:
a. Debabrata Ghosh et al. "Quantitative evaluation of image mosaicing in multiple scene categories." In Electro/Information Technology, 2012 IEEE International Conference on, pp. 1-6. IEEE, 2012.
5) It is unclear how many set of images were used for this work. The results may be different from one set to another set of images.
6) How the noise and blurriness affect the efficiency of the algorithm?
7) The conclusion is too short
8) The references have to include papers of authors that with aerial image mosaicking and contributed to this field.
a. Sangho et al. "Hierarchical multi-level image mosaicing for autonomous navigation of UAV." In Intelligent Robots and Computer Vision XXIX: Algorithms and Techniques, vol. 8301, p. 830116. International Society for Optics and Photonics, 2012.
b. D. Ghosh et al. "Robust spatial-domain based super-resolution mosaicing of CubeSat video frames: Algorithm and evaluation." Computer and Information Science 7, no. 2 (2014): 68.
Author Response
We would like to thank the reviewer for the constructive comments, which enable us to greatly improve the quality of our manuscript. We have provided a marked-up copy to highlight the changes in our revision. Next, we provide point-by-point responses to the review comments.
1) The paper needs some English editing
Line 1: Study should be replaced by Paper
Line 2: This second sentence should be revised
Line 16: What does it mean “good safety”?
Reply: Thanks for pointing out these problems. We have carefully proofread the paper several times. In particular, in Line 16, what we want to express is that it is safer to use drones to get images in certain dangerous places.
2) The authors state in the abstract that their method outperforms traditional methods. The Experimental Results they state that their method is compared to the traditional image mosaic method. They are inconsistencies in the authors’ statements. I also do not see any comparison in the paper.
Reply: Thanks for pointing out this problem. The traditional method we mentioned is Autostitch. In this paper, we compared our method with Autostitch, and compared it with some other advanced methods. We also described the experimental results in detail and made modifications in the paper.
3) I do not see any quantitative evaluation in the paper. The authors can refer to the following about metrics to evaluate their algorithm:
a. Debabrata Ghosh et al. "Quantitative evaluation of image mosaicing in multiple scene categories." In Electro/Information Technology, 2012 IEEE International Conference on, pp. 1-6. IEEE, 2012.
Reply: Thanks for pointing out this problem. In the introduction and summary sections of the paper, we all mentioned that our method is superior to the traditional method in quantitative. It is not embodied in the experimental results. We made some mistakes in writing. What we mainly want to express is that our experimental results are better than traditional methods and some advanced methods in qualitative. In particular, there are no quantitative indicators in all our comparative papers either, like Autostitch , SPHP ,AANAP ,etc. Nevertheless, we would like to thank the reviewer to provide us a reference about metrics. We have commented it in the related work as follows.
“Ghosh et al. also put forward some indicators for quantitative evaluation of image mosaicing in multiple scene categories [43], which also made up the defects in this aspect.”
4) The fourth question is missing in the original review comments.
5) It is unclear how many set of images were used for this work. The results may be different from one set to another set of images.
Reply: Thanks for pointing out this problem. In our revision, we have added several image sets of different types of scene for stitching, as shown in Figures 3-8. In these different types of scene, our method can consistently produce satisfying results. These experimental results make our method more convincing.
6) How the noise and blurriness affect the efficiency of the algorithm?
Reply: Thanks for pointing out this problem. As suggested, we have conducted relevant experiments to show the influence of noise and blur on our algorithm, as shown in Sec. 4.3 on Page 9. The experimental results show that our method is not affected by noise and blur.
7) The conclusion is too short
Reply: Thanks for pointing out this problem. According to your suggestion, we have rewritten the conclusion section. We also made some prospects for the future work to make up for the shortcomings of our methods. The conclusion is as follows.
“In this study, we propose an image mosaic method based on robust feature matching and a new bundle adjustment strategy. The use of robust feature matching makes the results more accurate and reduces the pressure of the subsequent steps of the system. When calculating the transformation relation between images, we differentiate the global homography matrix so that the new adjustment is only needed to blend the aligned image with the average intensity. Results show that our approach provides a more natural panorama with no visible parallax in the overlapping regions and the ghost effect produced by mosaicing is obviously reduced. Our system is more suitable for UAV images and has a wider application range than the traditional image mosaic method. Future research developments will improve the image effect in the case of large parallax, which can be performed by integrating seam-cut methods into stitching framework.”
8) The references have to include papers of authors that with aerial image mosaicking and contributed to this field.
a. Sangho et al. "Hierarchical multi-level image mosaicing for autonomous navigation of UAV." In Intelligent Robots and Computer Vision XXIX: Algorithms and Techniques, vol. 8301, p. 830116. International Society for Optics and Photonics, 2012.
b. D. Ghosh et al. "Robust spatial-domain based super-resolution mosaicing of CubeSat video frames: Algorithm and evaluation." Computer and Information Science 7, no. 2 (2014): 68.
Reply: Thanks for pointing out this problem. As suggested, we have added a section in the related work as follows.
“The theme of this paper is about the stitching of UAV images. In this part, we introduce the related work in aerial image mosaic.
Park et al. [41] proposed an algorithm for hierarchical multi-level image mosaicing for autonomous navigation of UAV. The algorithm prevents the error accumulation propagated along the frames, by incrementally building a long-duration mosaic on the fly which is hierarchically composed of short-duration mosaics. It fulfills the real-time processing requirements in autonomous navigation. In particular, the system can automatically adapt to the scene changes of the images to be spliced, and can automatically select more suitable methods for feature selection. This is an important point in the autonomous navigation of UAV. The proposed system is a causal system and suitable for real-time application. Ghosh et al. [42] proposed a spatial domain based super-resolution mosaicing system and its evaluation results. This algorithm combines image mosaicing algorithm and super-resolution-reconstruction algorithm, and has the characteristics of robustness and simple calculation. The author uses spatial domain based super resolution and makes it practical in real time applications like surveillance and remote sensing. Ghosh et al. also put forward some indicators for quantitative evaluation of image mosaicing in multiple scene categories [43], which also made up the defects in this aspect.”
Round 2
Reviewer 1 Report
In the revised version, Fig.4 and Fig.6 were newly added to demonstrate the stitching results. However, in Figures 4, 5, and 6, the image resolution of the stitching results in the last row (i.e., the proposed approach) are higher than the rest images (i.e., the results of the other state-of-the-art methods). The reviewer feels that it’s not really a fair qualitative comparison. Moreover, the sizes (or resolutions) of all the images are quite small, therefore, it is hard to compare the quality of different stitching results.
It is suggested to conduct a set of synthetic experiments to analyze how the mentioned challenges (such as non-rigid changes caused by ground relief, and parallax caused by changes in imaging viewpoints) affect the mosaicing result.
Most of the experiments demonstrate the mosaic results of stitching only two images. In order to generate wide field-of-view panoramic image, the algorithm should be able to handle multiple images captured from “arbitrary viewpoints”. Figure 8 shows an example of sequential images captured from left to right, but this is only one case. More experiments on stitching multiple images of arbitrary viewpoints should be provided to evaluate proposed bundle adjustment method.
In the feature matching experiment (Figure 1 and Table 1), there is still no information on how the ground truth was obtained. If this image pair is some benchmark provided by other author, please cite the source.
Lastly, some sentences should be revised in term of English.
Author Response
We would like to thank the reviewer for the constructive comments, which enable us to further improve the quality of our manuscript. We have provided a marked-up copy to highlight the changes in our revision. Next, we provide point-by-point responses to the review comments.
1) In the revised version, Fig. 4 and Fig. 6 were newly added to demonstrate the stitching results. However, in Figures 4, 5, and 6, the image resolution of the stitching results in the last row (i.e., the proposed approach) are higher than the rest images (i.e., the results of the other state-of-the-art methods). The reviewer feels that it’s not really a fair qualitative comparison. Moreover, the sizes (or resolutions) of all the images are quite small, therefore, it is hard to compare the quality of different stitching results.
Reply: Thanks for pointing out this issue. In our revision, we have changed the image resolution of the stitching results of different methods, so that all the resulting images have the same resolution. Moreover, we also have zoomed in on the area in red boxes to clearly compare the results. The red boxes highlight the ghostly parts. Please refer to the Figures 4, 5, and 6.
2) It is suggested to conduct a set of synthetic experiments to analyze how the mentioned challenges (such as non-rigid changes caused by ground relief, and parallax caused by changes in imaging viewpoints) affect the mosaicing result.
Reply: Thanks for this suggestion. As for this problem, we did not describe it in our previous paper. All the above mentioned challenges will bring ghost effects to the stitching results, and our method is mainly to eliminate these ghost effects. In our revision, we have added a new section to discuss this issue, please refer to Sec. 4.3 “The Effect of Nonrigid Changes and Parallax”. Combined with our experimental results, we analyze how the mentioned challenges (such as non-rigid changes caused by ground relief, and parallax caused by changes in imaging viewpoints) affect the mosaicing result in the revised paper.
3) Most of the experiments demonstrate the mosaic results of stitching only two images. In order to generate wide field-of-view panoramic image, the algorithm should be able to handle multiple images captured from “arbitrary viewpoints”. Figure 8 shows an example of sequential images captured from left to right, but this is only one case. More experiments on stitching multiple images of arbitrary viewpoints should be provided to evaluate proposed bundle adjustment method.
Reply: Thanks for pointing out this problem. Our datasets were obtained by autonomous flight of UAV. In order to obtain images in the field, track planning of UAV is usually adopted. Drones usually fly in a straight line, and the camera angle is fixed, so the images we get are all in a sequence. In order to verify the robustness of our algorithm for multiple images of arbitrary viewpoints, we conduct new experiments on a publicly available dataset involving such type of images (Available at: https://www.sensefly.com/drones/example-datasets.html). The experimental results are shown in Fig. 10 and Fig. 11.
4) In the feature matching experiment (Figure 1 and Table 1), there is still no information on how the ground truth was obtained. If this image pair is some benchmark provided by other author, please cite the source.
Reply: Thanks for pointing out this problem. In our revision, we have cited the source (i.e., Ref. [31]), where the ground truth feature matches are supplied by the dataset. Please refer to Sec. 4.1 on Page 7.
5) Lastly, some sentences should be revised in term of English.
Reply: Thanks for pointing out this problem. We have asked for professional English editing service to polish our paper.
Reviewer 2 Report
The authors have addressed all the comments.
Author Response
We would like to thank the reviewer for the positive feedback.
Round 3
Reviewer 1 Report
no